# Successful Genetic Screening and Creating Awareness of Familial Hypercholesterolemia and Other Heritable Dyslipidemias in the Netherlands

**DOI:** 10.3390/genes12081168

**Published:** 2021-07-29

**Authors:** Linda C. Zuurbier, Joep C. Defesche, Albert Wiegman

**Affiliations:** 1Department of Human Genetics, Amsterdam University Medical Center, 1105 AZ Amsterdam, The Netherlands; l.c.zuurbier@amsterdamumc.nl (L.C.Z.); j.defesche@amsterdamumc.nl (J.C.D.); 2Department of Paediatrics, Amsterdam University Medical Center, 1105 AZ Amsterdam, The Netherlands

**Keywords:** dyslipidemia, familial hypercholesterolemia, cholesterol, lipids, genetic screening

## Abstract

The genetic screening program for familial hypercholesterolemia (FH) in the Netherlands, which was embraced by the Dutch Ministry of Health from 1994 to 2014, has led to twenty years of identification of at least 1500 FH cases per year. Although funding by the government was terminated in 2014, the approach had proven its effectiveness and had built the foundation for the development of more sophisticated diagnostic tools, clinical collaborations, and new molecular-based treatments for FH patients. As such, the community was driven to continue the program, insurance companies were convinced to collaborate, and multiple approaches were launched to find new index cases with FH. Additionally, the screening was extended, now also including other heritable dyslipidemias. For this purpose, a diagnostic next-generation sequencing (NGS) panel was developed, which not only comprised the culprit *LDLR*, *APOB*, and *PCSK9* genes, but also 24 other genes that are causally associated with genetic dyslipidemias. Moreover, the NGS technique enabled further optimization by including pharmacogenomic genes in the panel. Using such a panel, more patients that are prone to cardiovascular diseases are being identified nowadays and receive more personalized treatment. Moreover, the NGS output teaches us more and more about the dyslipidemic landscape that is less straightforward than we originally thought. Still, continuous progress is being made that underlines the strength of genetics in dyslipidemia, such as discovery of alternative genomic pathogenic mechanisms of disease development and polygenic contribution.

## 1. Non-Communicable Diseases–Cardiovascular Diseases

The third World Health Organization global status report on non-communicable diseases (NCDs) 2018, states that a total of 41 million deaths each year (71% of all deaths) were due to NCDs, and cardiovascular diseases account for most NCD deaths, namely 17.9 million annually [1]. Eurostat data published in July 2018 states that: “Cardiovascular diseases (CVDs) are the leading cause of death in the European Union. CVDs cover a broad group of medical problems that affect the circulatory system (the heart, blood vessels and arteries) often resulting from atherosclerosis - abnormal build-up of plaque made of cholesterol - deposited on the inside walls of arteries. Some of the most common diseases that affect the circulatory system include ischemic heart disease (heart attacks) and cerebrovascular diseases (strokes)” [2]. An important risk factor for developing ischemic heart disease through atherosclerosis is hypercholesterolemia, a lipid disorder leading to a serious accumulation of low-density cholesterol (LDL-c) in the blood flow. Hypercholesterolemia can occur secondary to other diseases, medication, or lifestyle, or primary as a result of pathogenic variants in the DNA, a disease that is known as familial hypercholesterolemia (FH). Patients with a genetic background for FH are at a considerably higher risk for coronary heart disease, compared to those with an idiopathic or secondary cause for FH, and may be treated more aggressively [3]. Next to FH, other lipid disorders exist, which are related to abnormal concentrations of different lipoprotein particles, and can also occur as both primary and secondary. These include high levels of lipoproteins such as high-density lipoprotein (HDL) particles (hyperalphalipoproteinemia) or very-high-density lipoprotein (VLDL) particles (hypertriglyceridemia, hyperchylomicronemia, or dysbetalipoproteinemia), low levels of lipoprotein particles (hypolipoproteinemia’s), and atypical levels of sterols or chylomicrons (sitosterolemia, cerebrotendineous xanthomatosis, chylomicron retention disease). These lipoprotein disorders altogether are grouped as dyslipidemias; most are clinically relevant because they increase CVD risk, but others involve low LDL-c and/or triglyceride levels, and are cardioprotective. The relevance of high HDL levels in relation to CVD are still disputable. 

## 2. Familial Hypercholesterolemia

FH is the most common and serious monogenic dyslipidemia that leads to premature CVD, but despite major advances in scientific and clinical knowledge about the condition, most cases remain undetected or inadequately treated [4]. In cases with FH, plasma levels of LDL-c are severely elevated from birth onward, putting them at high risk of premature CVD [5]. CVD might already occur in the first or second decade of life in the rare form of homozygosity (~3 per million) [6], whereas in the frequent form of heterozygosity (~1 in 300) [7,8] half of untreated heterozygous FH (heFH) ends in premature CVD before the age of 55. Worldwide, one baby with heFH is born every minute, and one baby with homozygous FH (hoFH) every day [9], but fortunately, FH is well detectable and treatable, and the above shows there is need to detect and treat it [10].

Generally, FH can be detected by measuring plasma LDL-c levels, but, if not performed properly with at least a diet interval of three months, it is a less reliable diagnostic tool. Genetic testing, on the other hand, introduces several positive effects after a pathogenic variant in an index case has been found. At first, the awareness increases that high LDL-c is present from birth onwards. Second, the awareness that the pathogenic factor is passed on to the next generation develops. Third, for family members, there is an early opportunity to get certainty about the variant being present or absent. Fourth, physicians get a better understanding of the severity of all types of pathogenic variants and of who, how, and when to treat. Finally, and not hypothetically, by testing DNA before potential parenthood, knowing one’s defects, homozygosity in future children can be prevented. In the whole process of DNA testing, it is pivotal to involve insurance companies to avoid higher insurance premiums for individuals at risk identified by presymptomatic DNA testing, since they will be treated early in life and will gain decades of healthy life.

Currently, the overall mean age in the Netherlands is 82 years, but in heFH this is only 61 years [11]. If nothing is done, a loss of 21 healthy life-years occurs. Relatively easily accessible and cheap treatment options are available for FH patients. The most broadly prescribed are statins, which are orally available inhibitors of cholesterol synthesis. Multiple studies over the last years have shown not only the benefits of lower LDL-c levels, but also the benefits of longer low levels [12,13,14]. This underlines the importance of early identification of FH to lower LDL-c levels by treatment at a young age. Even treatment at a later age still shows a significant lowering in intrinsic LDL-c levels after treatment, and with this, an extension of the lifetime before a first CVD event occurs (Figure 1) [15,16]. The necessity and benefits of early detection and early treatment have recently been demonstrated by a twenty-year follow up study in heFH patients that started treatment in childhood (Figure 2) [17]. 

## 3. Lipids and Dyslipidemia

The cholesterol mechanism involves different lipoprotein particles that transport cholesterol through the blood circulation towards tissues. These particles differ in size and composition according to phospholipids, apolipoproteins, and cholesterol and triglycerides content; therefore, each particle has a different effect on plaque development in the arteries. 

LDL particles transport cholesterol to the peripheral tissue and are small enough to cross the vascular endothelium. The LDL receptor (LDLR) allows an LDL particle to be taken up by cells from the blood. Newly synthesized receptors migrate to the cell surface where they can bind LDL. However, within the tissue, the LDL particles become oxidized (OxLDL), and OxLDL may be integrally involved in plaque-forming and -destabilization [18]. Although cholesterol is an essential component of cell membranes and is also the precursor for steroid hormones and vitamin D, there is no obvious explanation for very high levels of LDL. Plasma LDL-c in excess of 3.0 mmol/L (116 mg/dL) is regarded as elevated and should not be exceeded for a longer period [19]. In contrast, low LDL-c levels are cardioprotective, although severely decreased levels can result in problems related to fat absorption or fat-soluble vitamins, for example steatorrhea, vitamin deficiency, or failure to thrive in early childhood [20].

HDL particles are involved in reverse cholesterol transport and deliver cholesterol to the liver for catabolism. HDL is believed to have a protective role against atherosclerosis, also because it protects LDL against atherogenic oxidative modification [21]. Consequently, low levels of HDL cholesterol (HDL-c) are associated with an increased risk of CHD [22,23]. In contrast, high levels of HDL-c were first assumed to be cardioprotective [24]. This was corroborated in 2016, when a functional variant was reported in the *SCARB1* HDL receptor that was associated with increased HDL-c levels and a higher risk of CHD [25]. Nonetheless, the effect of high HDL-c levels on the incidence of CHD is currently disputable based on a clinical trial with the HDL-increasing CETP inhibitor torcetrapib that was associated with a higher incidence of death [26,27]. Additionally, Mendelian randomization studies could not support a causal relationship between HDL-c and CVD risk [27].

Triglycerides constitute our major energy store. Serum triglycerides are usually considered elevated if they exceed 2.0 mmol/L (150 mg/dL) [28]. Incidentally, elevated triglyceride levels are relatively harmless and rapidly normalize, but prolonged high serum triglycerides can be detrimental and are often seen in obesity in combination with low HDL [29]. Severe hypertriglyceridemia predominantly has a primary origin and needs treatment to avoid acute pancreatitis [28]. The association of hypertriglyceridemia with coronary heart disease (CHD) is complex and under constant examination. More clear is the knowledge on low triglyceride levels, which are associated with a lower risk of CVD [30,31,32].

In our genetics laboratory and pediatric lipid clinic, the focus has been primarily on high LDL-c levels (FH), and still the majority of identified cases are those with FH. We have known for years now that pathogenic variants in *LDLR*, *APOB*, and *PCSK9* genes are the major cause of FH. Furthermore, we know that mutated genes or even specific variants are associated with a particular average height of LDL-c, such as that LDLR null/deficient alleles (e.g., CNVs, nonsense, and splice variants) in general result in very high LDL-c levels (~6 mmol/L) and LDLR defective alleles (most missense variants) in somewhat lower levels (~4.5 mmol/L) [33]. Additionally, *APOB* variants result in milder LDL-c increases compared to *LDLR* variants [3,34]. In addition to this, recently, Khera and colleagues elegantly showed that having an FH variant, independent of LDL-c level, already significantly increased the risk for CHD [3]. This is probably due to a lifelong burden of increased LDL-c levels. They also showed that, although this was true for all LDL-c level strata, this difference became more apparent when LDL-c levels increased; in the highest LDL-c stratum, the odds increased 7.7-fold for those without an FH variant and 25.8-fold for those with a variant, compared to the reference group without a variant and LDL-c levels below 130 mg/dL (3.4 mmol/L). These are challenging data that support the need for pro-active genetic screening for new index patients and family members.

Until now, most research and screening efforts have been focused on FH, despite the association of other forms of dyslipidemia with CVD. Fortunately, this area is slowly shifting as evolved laboratory techniques now allow us to more easily detect variants in many genes at once. With early identification, patients with other dyslipidemia can also be prevented from arterial plaque formation and subsequent myocardial infarction.

## 4. The Dutch FH Screening Project

The Netherlands is one of the few countries that is very active in FH screening. This could not have been realized without the involvement of the government as of the year 1994, who made it possible to perform pro-active home visits by specialized nurses (the so called genetic field workers) for genetic family screening, from the age of six years on. The program has evolved from a regional pilot research project funded by the Dutch Ministry of Health to full nationwide population screening, based on promising results in detecting and treating FH in adults. The cascade screening program was approved by the National Ethics Committee. Insurance companies promised not to charge penalties for having a mutation. On the contrary, they encouraged early detection and prevention. Privacy and other legal protection issues were well arranged in co-operation with the Ministry of Health. 

The estimated prevalence of FH in 1994 (1:500), combined with 15 million inhabitants, resulted in an expected number of 30,000 FH patients. It was estimated that most of them would be traced in a time frame of twenty years [35]. The results of the first five years of cascade screening for FH in the Netherlands were published in 2001 [36]. The screening program aimed to establish the feasibility of an active family screening supported by DNA diagnostics and to assess whether the active identification of cases with FH would lead to improvements in the extent of preventive care. A further aim was to assess the specificity and sensitivity of cholesterol measurement by comparing molecular diagnosis with cholesterol measurements in families in which a pathogenic variant in the *LDLR* gene had been detected. Participation rates in this program were high, possibly influenced by reassurances that at least 50% of participants tested were not at risk, would not develop the disorder, and would not pass it to their children. By contrast, screening programs for disorders for which no treatment is available have much lower participation rates [37]. At the time of examination, only 39% of adult patients with FH initially identified were receiving some form of cholesterol-lowering treatment—one year later this percentage had risen to 93% [36]. The one-year data was judged as being an important public health milestone. In Dutch cases with FH, more than 1000 different pathogenic *LDLR* variants were identified, and accounted for about 80% of patients of Dutch origin. By the beginning of 2014, when the program stopped, 64,171 subjects had undergone genetic testing for FH. Of these, 26,232 (40.8%) were heFH mutation carriers and 37,939 (59.2%) were unaffected relatives [35]. Together with the 4000 index cases, the program has led to the identification of over 30,000 FH cases.

Despite the successful program, the funding by the government has been discontinued. In 2014, however, the estimated prevalence of FH rose to 1:244. This, combined with an increased number of 17 million inhabitants, resulted in an expected number of 70,000 FH patients [38]. This probably means that 40,000 cases have yet to be found. After the termination of the government-funded program in 2014, the number of newly identified FH cases dramatically dropped. While active cascade screening found at least 1500 FH cases each year, in 2015, only 360 new FH family members were identified, and numbers steadily rose towards 758 and 639 in 2019 and 2020, respectively. However, the program had proven its effectiveness and it motivated clinicians to voluntarily continue the screening program without funding from the government. Instead, the accomplishment of the program convinced stakeholders such as the Netherlands Heart Foundation and insurance companies to cover costs for further active screening and to dismiss negative consequences for patients regarding life insurance. 

For this purpose, a non-profit organization (LEEFH foundation) was launched, which is now involved in the national coordination of family screening for FH as well as in stimulating individual genetic screening for those at risk. The foundation plays a facilitating role in a growing voluntary FH network of 29 Dutch hospitals that have a crucial role in the regional communication and cooperation with general practitioners in stimulating family screening. In addition to this, the LEEFH foundation is encouraging and actively participating in projects that aim to get yet uncovered FH cases in sight. For example, a number of hospitals are starting to use electronic health records to facilitate continuous detection of FH [39]. Other forms of screening are also being explored, such as reverse cascade screening, selective genetic screening of those with LDL-c levels >95th percentile, and universal screenings of children during routine visits. Molecular biologists assert that detection in childhood instead of adulthood increases the chance of discovering pathogenic variants, because at a young age there are few secondary causes that interfere with a clear cut diagnosis [40]. Each new case can be followed up with child–parent testing or reverse cascade screening (Figure 3). This turns out to be effective because, for every child with a molecular proven FH variant, one of both parents is affected as well. 

## 5. Screening beyond FH; Inclusion of 24 Additional Genes of the Cholesterol Metabolism

More than twenty years ago, genetic screening for dyslipidemia was restricted to FH by Sanger sequencing of the *LDLR* promoter and coding region and exon 26 and 29 of *APOB*, the only two *APOB* regions known for pathogenic FH variants. Later on, CNV analysis using MLPA (multiplex ligation-dependent probe amplification) became available and was applied to every individual that had significantly elevated LDL-c levels, but lacked pathogenic variants. The panel was expanded in 2003, when *PCSK9* gain-of-function (GOF) variants were discovered as a third causal mechanism for FH [41].

*PCSK9* is a member of the proprotein convertase family and consists of a signal peptide domain, prodomain, a catalytic domain, and a carboxy-terminal domain that is needed for PCSK9 localization. Interaction with the LDLR protein targets LDLR for degradation by the lysosomes and thereby prevents recirculation of LDLR to the cell membrane. To date, ~30 GOF variants and another ~30 LOF variants are known to be functional, and they are spread over the whole gene [42]. In the Netherlands, those *PCSK9* GOF variants were detected in ~5% of FH patients, compared to ~20% APOB variants and ~75% LDLR variants.

Whereas many laboratories worldwide still screen for a limited number of genes or variants only, NGS was introduced in the Netherlands in 2016 as new diagnostic tool for dyslipidemia. With this approach, genes beyond *LDLR*, *APOB*, and *PCSK9* could be included in the screening. As such, FH was not the only focus anymore, and patients with other heritable lipid disorders could also be identified and receive adequate treatment. At the moment, genetic testing for dyslipidemia in the Netherlands includes causality genes for familial dysbetalipoproteinemia, hypertriglyceridemia, hyperalphalipoproteinemia, hypolipoproteinemia, and rare diseases sitosterolemia, recessive hypercholesterolemia, chylomicron retention disease, cerebrotendineous xanthomatosis, and cholesteryl ester storage disease (or Wolman disease in specific cases). The Dutch NGS panel currently includes 27 genes, for which a clear relation between variants and dyslipidemia has been established (Table 1) and allows copy number variant (CNV) analysis of all genes in the panel. The panel deliberately excludes genes for which causality is not rigorously proven to avoid the over reporting of variants with unclear meaning (variants of unknown significance, VOUS) that can unnecessarily stress patients. The 24 additional genes comprise causal genes for rare lipid disorders, such as cholesteryl ester storage disease/Wolman disease (*LIPA* variants) and cerebrotendineous xanthomatosis (*CYP27A1* variants), and those that are causally involved in the pathogenesis of dyslipidemia and encode major players of the cholesterol metabolism. 

Panel-included key genes that encode proteins related to hypertriglyceridemia are *LPL* (lipoprotein lipase), *APOC2* (apolipoprotein C2), *APOA5* (apolipoprotein A5), *LIPC* (Lipase C), *GPIHBP1* (glycosylphosphatidylinositol-anchored high-density lipoprotein binding protein 1), and *LMF1* (lipase maturation factor 1), and, later, *GPD1* (glycerol-3-phosphate dehydrogenase 1) was also included. LPL is an enzyme important for the hydrolysis of triglyceride-rich lipoproteins (TRLs) for storage by the adipose tissue or energy supply by the muscles. TRLs are very low-density lipoproteins (VLDL) produced by the liver and chylomicrons that are generated post-prandial in the intestine and include dietary triglycerides. Hydrolysis of these TLR particles (lipolysis) leads to the formation of smaller and denser remnants (intermediate LDL, LDL, and remnant chylomicrons) and the release of free fatty acids that can be absorbed by the adipose or muscle tissue. After hydrolysis, chylomicron remnants are catabolized in the liver. VLDL remnants, on the other hand, first supply peripheral tissues with cholesterol and are thereafter catabolized by the liver. Clearance of remnants by the liver is mediated by apolipoprotein E (APOE) and APOB on lipoproteins and LDL-receptors (LDLR) expressed on hepatocytes [43]. 

LPL is present on the cell membrane of the endothelial in different tissues–mainly adipose and muscle tissue and the heart. Its activity is regulated by various proteins, depending on the fasting state. When active, LPL needs anchoring in the cell endothelial membrane, which is mediated by GPIHBP1. APOC2 and APOA5 are present on lipoprotein particles and are LPL activators, whereas APOC3 and ANGPTLs are an inhibitors of LPL activity. LMF1 is essential for the post-translational modification of LPL, which is necessary for LPL catalytic activity and proper secretion. Hepatic lipase (HL, encoded by the *LIPC* gene) has a similar function as LPL but mediates the hydrolysis of IDL to LDL to small LDL and of HDL to smaller HDL particles. 

Loss of LPL activity by genetic variants, in either LPL itself or in its supporting proteins, results in prolonged presence of VLDL in the circulation, which is related to increased CVD risk and increased risk for acute pancreatitis [44]. Loss of function of those proteins is not necessarily pathogenic, as lipolysis can still be managed under normal conditions. However, when the supply of endogenous or exogenous triglyceride increases, for example by high fat intake, the system is not able to maintain the processing of VLDL, IDL, LDL, and chylomicrons anymore, and a hypertriglyceridemia arises. In contrast, a hypolipidemia can occur by LOF variants in genes encoding proteins that are involved in the production of VLDL particles, such as *APOB* or microsomal TG transfer protein (*MTP*). It looks like that these variants are associated with a lower risk for CVD. Additionally, particular variants in *ANGPTL3* are associated with hypolipoproteinemia.

Peripheral cells that are not able to catabolize cholesterol use a reverse cholesterol transport mechanism in which functional HDL particles are crucial. Primarily, mature HDL particles re-transport cholesterol back to the liver, where it binds, among others, the receptor scavenger–receptor class B type 1 (SCARB1 or SR-B1). In addition, it can also exchange its cholesterol content with triglycerides present in other lipoproteins (chylomicrons, VLDL and LDL). This process is mediated by cholesteryl ester transfer protein (CETP). These HDL-triglycerides are subsequently hydrolyzed by HL, thereby releasing free fatty acids and initiating the subsequent breakdown of HDL particles. Phospholipids of HDL are hydrolyzed by endothelial lipases (EL, *LIPG*). On the other hand, the cholesterol that is transferred to chylomicrons, VLDL, and LDL is reprocessed in the cholesterol metabolism. Cholesterol in the liver ends up in the bile, involving ABCG5 and ABCG8 proteins for transport. These ABCG members also transport plant sterols towards the bile and the complete loss of function of these ABCG proteins results in sitosterolemia. 

HDL is a particle that is constructed by apolipoprotein A1 (Apo A-1) and filled with excessively present cholesterol and phospholipids from peripheral cells by the ABCA1 transporter protein to form a premature nascent HDL molecule. HDL molecules mature to large HDL by transferring free cholesterol from the surface of HDL to the inside. This relocation needs esterification of free cholesterol, a process that depends on lecithin: cholesterol acyltransferase (LCAT). A loss of proteins that are indispensable for creating and maturing HDL, such as ABCA1, Apo A-1, and LCAT, leads to low HDL-c levels in the circulation, whereas a loss of those involved in HDL catabolism result in high HDL levels. 

Using the 29-gene panel, in the Netherlands, the most frequently found variants that lead to a dysregulated cholesterol metabolism are those resulting in FH (~16%) [33]. Around 4% of cases have CNVs, which are predominantly in *LDLR*. In some genes, both loss-of-function (LOF) and gain-of-function (GOF) variants can occur, resulting in opposite phenotypes. For example, *PCSK9* variants can cause hypercholesterolemia or hypocholesterolemia [38,39]. Similar mechanisms could be hypothesized for other dyslipidemic genes, such as *LDLR*, *APOC3* or *ANGPTL3*. Interestingly, a first report of a GOF variant in *LDLR* comprising a 3’UTR deletion has recently been published in a family with 74% lower LDL-c levels compared to non-carriers [40]. The authors suggest that the variant deletes negative regulatory elements such as miRNA-binding sites. Similarly, in 2017, the variant Gln38Lys in *APOC3* has been reported as a possible GOF variant causing hypertriglyceridemia, instead of the well-known LOF mechanisms leading to hypolipoproteinemia [41]. In addition, LOF variants in *APOB* can cause both hypercholesterolemia or hypobetalipoproteinemia, depending on the type and location of the variant in the gene; missense variants in exon 26 of *APOB* or nonsense variants in exon 29 lead to hypercholesterolemia, whereas nonsense variants in the first half of *APOB* cause hypobetalipoproteinemia [42,43]. *APOB* is a large gene containing many missense variants. Only a few variants are adequately proven to cause hypercholesterolemia. 

Interestingly, the multilocus screening approach also easily allowed the addition of pharmacogenetic genes in the panel, such as *SLCO1B1* (solute carrier organic anion transporter). *SLCO1B1* encodes an organic anion-transporting polypeptide 1B1 (OATP1B1) transporter protein, which facilitates the clearance of endogenous compounds such as statins from the circulation. Statins are 3-hydroxy-3-methylglutaryl coenzyme A (HMG-CoA) reductase inhibitors and form the backbone of treatment for hypercholesterolemia cases. When OATP1B1 transporters are less or not functional, statins are retained in the circulation, which can lead to muscle toxicity and thereby statin-associated muscle symptoms (SAMS). SAMS occur in 10–33% of patients on statins in observational studies, and range from mild in most individuals to serious in some individuals. In a genome-wide association study in 2008 [44], it was shown that the variant c.521T>C, p.(Val174Ala) in *SLCO1B1* (rs4149056) was associated with increased plasma levels of statins and with odds ratios of 4.5 (heterozygous) and 16.9 (homozygous) for developing SAMS in individuals on simvastatin. These results have been confirmed in subsequent studies [45,46,47]. Worldwide, 13% of individuals (GnomAD database [48]) carry this variant, and therapeutic recommendations have been published when prescribing simvastatin in individuals with this variant [49]. More variants have been associated with reduced OATP1B1 transporter activity, which, for example, has been shown for the transport of methotrexate in children with acute lymphoblastic leukemia or bilirubin in case of Rotor syndrome, suggesting that more variants in *SLCO1B1* can affect statin clearance and are thereby able to evoke SAMS. In Dutch individuals that were analyzed with the NGS panel, multiple other variants as well as whole gene deletions have been observed. SAMS frequently led to non-adherence or discontinuation of statins, which has been associated with an increase in cardiovascular events or death [50,51]. In 2018, a randomized study was performed using genotyping for re-initiation statins, in which it was shown that genotyping resulted in a significantly larger number of individuals that reinitiated on statins [52].

## 6. Lessons from NGS Dyslipidemia Panel

The introduction of NGS led to a significant increase in the number of identified index cases with genetic dyslipidemia. In the Netherlands, over 2000 cases are now genotyped each year. Whereas, in the first decade of the 20th century, only 15% of diagnostically screened cases were genetically confirmed with FH using conventional techniques [45], a growth towards 30% of heritable dyslipidemia cases was noticed using the broader screening approach. It was observed that the relative number of index cases with FH remained stable over the years, and that the additional index cases represented cases with non-FH dyslipidemia [34]. The majority of these non-FH cases were genetically diagnosed with hypertriglyceridemia, dysbetalipoproteinemia, or hypoalphalipoproteinemia. Interestingly, a major proportion of these cases was initially clinically diagnosed with FH, indicating that it is difficult to make a clinical distinction based on lipid profiles only, and underlines the significance of genetic testing.

Many of those cases were explained by the presence of multiple functional variants in different genes leading to a blend of two or even three dyslipidemias, or dyslipidemias with a contrary effect on lipid levels. For example, in cases with FH, additional hypobetalipoproteinemia variants have been found [46]. These additional variants can explain normal LDL-c levels in FH cases and large differences in LDL-c levels within an FH family. Furthermore, mimicking FH variants have been identified. For example, *ABCG5/ABCG8* and *APOE* variants, also called minor FH genes, are generally known to cause sitosterolemia (recessive) or dysbetalipoproteinemia (dominant), respectively, but interestingly, an association with hypercholesterolemia has also been published for some heterozygous variants in both genes, although with a less severe phenotype [47,48,49,50,51]. Regarding *APOE*, in a French cohort of *LDLR/APOB/PCSK9*-negative hypercholesterolemia cases with normal triglyceride levels, three out of 229 (1.3%) carried an APOE variant, of which 2 were located in the APOE–LDLR-binding domain [49]. Furthermore, in 153 other hypercholesterolemia probands, three *APOE* variants were detected, of which the p.(Leu167del) variant was found to segregate within a family with hypercholesterolemia [53]. Pathogenicity of these variants was supported by in vitro LDL kinetic studies and in silico structural prediction and this observation was confirmed in later studies [52,54]. Additionally, within the Dutch NGS cohort, a substantial number of *APOE* and heterozygous *ABCG5/ABCG8* variants have been noticed in clinical FH patients [47].

In addition to these contradicting and mimicking phenotypes, variants in the *LIPC* gene seem to evoke two different phenotypes. *LIPC* encodes the enzyme hepatic lipase and variants in this gene result in hepatic deficiency. Based on data from the Dutch NGS cohort and data of others, genetic variants in *LIPC* as well as clinical hepatic lipase protein deficiency are known to associate with high HDL-c levels in some cases and high triglyceride levels in other cases [53,55,56,57]. More specifically, of those patients with *LIPC* variants, we observed HDL-c levels >95th percentile in seven individuals (of which one homozygous) and severely (> 10 mmol/L) elevated triglyceride levels in 15 individuals (one homozygous). Interestingly, not all cases with those variants showed these phenotypes, suggesting the need for additional factors, and all but one of the 13 different variants that were associated with triglyceride levels were located in exon 4, 5 or 6. It seems there is a yet unclear interaction between *LIPC* variants and high HDL-c or triglyceride levels, and it would be very interesting to further explore this phenomenon.

## 7. Heritable Hypertriglyceridemia

The most frequent non-FH variants that we observe in our laboratory in cases of dyslipidemia are those in genes involved in triglyceride regulation. These include dominant variants in *LPL*, *APOE*, *APOA5*, and *LIPC* and recessive variants in *APOC2*, *GPI-HBP1*, *LMF1, APOE*, and *GPD1*. In our NGS cohort, most pathogenic variants are located in *LPL*, *APOA5*, and *APOC2*. It is disputable what the role of some of these variants in disease pathogenesis is, as many of these variants only provoke dyslipidemia when secondary factors are involved, such as alcohol, diet, hormones, and other medications and secondary diseases such as diabetes and hypothyroidism [58,59]. Nevertheless, hypertriglyceridemia, which is frequently accompanied by lower HDL-c levels, can lead to serious health issues such as pancreatitis, and we frequently notice extremely high triglyceride levels in individuals with heterozygous functional variants in these genes. We therefore believe that these genes are indispensable in the screening process, and that the impact of these variants is frequently underestimated due to the absence of high triglyceride levels at the moment of measurement, as a consequence of temporarily elevated levels or the lack of secondary provoking factors. However, being aware of carrying such a variant can motivate individuals to adhere their healthy lifestyle. 

In our NGS data, it seems that some variants are more sensitive to lifestyle than others to provoke a phenotype. For example, of the 25 different lower frequency variants we detected in *LPL* in 36 individuals, 69% had hypertriglyceridemia, of which most (*n* = 22) had levels >10 mmol/L or the clinical indication of hypertriglyceridemia, and three had levels between 5 and 10 mmol/L. The remaining individuals had normal or unknown triglyceride levels. However, those numbers were lower for the higher frequency variants p.(Asp36Asn), p.(Val96Leu), p.(Gly215Glu), and p.(Asn318Ser). Regarding p.(Val96Leu), only 39% of cases had triglyceride levels above 10 mmol/L and 5% between 5 and 10 mmol/L. This observation could also be a result of a selection bias in our cohort. Nevertheless, a still substantial number of cases presented with extremely elevated triglyceride levels, and it is, therefore, important to actively perform cascade screenings in all cases with (lower frequency) *LPL* variants, as is done for FH, to prevent comorbidities by avoiding negative life style habits. As these provoking lifestyle factors are common and increasing in the general population, we feel the need to stress the importance of identifying these variants in the population.

## 8. Future Directions

Although the introduction of NGS has led to great advances, many dyslipidemic cases are still left undiagnosed because pathogenic variants are lacking [34]. In severe clinical FH cases, including Dutch patients with a modified definite Dutch Lipid Clinic Network criteria score of definite FH, a pathogenic FH variant was absent in 50–60% of cases [34,60] and comparable numbers were noted in other studies [61,62]. Future research may reveal new candidate genes for dyslipidemia or causal mechanisms in yet poorly explored regions of known dyslipidemic genes, such as deep intronic regions affecting regulatory elements like enhancer or boundary elements or affecting pre-mRNA splicing. In this light, using a screening algorithm, we have recently discovered a deep intronic variant c.2141-218G>A in intron 14 of *LDLR* that resulted in inclusion of a pseudo-exon in the LDLR mRNA presumably leading to truncated LDLR proteins, and this variant segregated with FH in a small family [63]. 

Another step forward in identifying more FH cases would be the development of functional tests to interpret the many variants of unknown significance (VOUS). In this context, segregation analysis must also be simulated. For this, close communication between physicians and molecular geneticists about the data is indispensable.

On the other hand, there are reasons to believe that increased LDL-c levels can have a polygenic origin, meaning that high frequency variants resulting in a small effect in LDL-c level alone, altogether can have a substantial impact. Such changes are captured in a polygenic risk score. So far, multiple polygenic risk scores have been developed for increased LDL-c levels using population data, including scores based on only six SNPs or based on hundreds of SNPs; however, these still need proven validation, clinical validity, and utility in hypercholesterolemia [64,65,66]. Such scores may explain high LDL-c levels in an individual or explain variance in LDL-c levels within FH families, but it is still debatable whether they are clinically actionable. Furthermore, presymptomatic cascade screening is not meaningful in these cases, as the combination of SNPs is not inherited in a Mendelian fashion and diluted in next generations. 

Alternatively, the role of lipoprotein(a) (Lp(a)) is a very interesting field for further exploration, as this lipoprotein is very similar to the LDL particle, and the cholesterol in Lp(a) is partly measured as LDL-cholesterol. Individuals with elevated Lp(a) levels have a significantly increased risk for CVD that is similar to that of FH patients. An association between Lp(a) levels and two SNPs in the *LPA* gene, encoding for Lipoprotein(a), the main protein constituent of Lp(a)), has been observed [67,68,69].

## 9. Discovery of Genes Led to Therapeutic Interventions

In the past, the discovery of new molecular mechanisms and genetic variants that cause FH has also led to the development of effective therapeutic targets. One of the most well-known targets is *PCSK9* (proprotein convertase subtilisin/kexin type 9). *PSCK9* is predominantly expressed in hepatocytes and promotes the degradation of LDL-receptor proteins by the lysosome, thereby preventing the elimination of LDL-c from blood circulation [70].

At first, linkage analysis revealed *PCSK9* as a candidate gene for hypercholesterolemia in two families in which *LDLR* and *APOB* variants were absent [41]. These families presented with the same c.625T>A *PCSK9* variant, and subsequent sequencing of 22 probands also revealed a third family with segregation of a *PCSK9* variant. A few years later, LOF variants in human *PCSK9* were discovered to be associated with very low LDL levels and fewer cardiovascular events [71,72,73]. This observation raised the idea of therapeutically inhibiting PCSK9 to attenuate LDL-c levels in hypercholesterolemic cases. For this purpose, monoclonal antibodies were developed, which blocked the function of PCSK9, and these were approved in 2015 [74]. The antibodies interfere with the interaction between PCSK9 and LDL-receptors. Recently, GalNAc-conjugated small interference RNAs (siRNAs) have also been developed, targeting only hepatocytes and inducing degradation of PCSK9 mRNA by an endogenous mechanism [75,76]. Currently, monoclonal PCSK9 antibody injections are prescribed in cases with an inadequate treatment response to statins or in those that are statin intolerant due to development of SAMS. 

A similar development recently took place for angiopoietin-like 3 (*ANGPTL3*). The ANGPTL3 protein inhibits the enzymes lipoprotein lipase (LPL) and endothelial lipase. LOF mutations in this gene associate with lower levels of triglycerides and LDL-c and fewer cardiovascular events [77]. This inspired researchers to develop pharmacological inhibitors of ANGPTL3. In this context, monoclonal antibodies as well as antisense oligonucleotides against ANGPTL3 proteins or mRNA, respectively, had been designed, and in vitro experiments proved the effectiveness for total cholesterol, triglycerides and LDL-c for both compounds [78]. Whereas ANGPTL3 siRNAs are still in phase one studies, monoclonal antibodies showed in 65 homozygous FH patients who did not achieve target LDL a similar positive effect as observed in vitro [79,80].

Additionally, the discovery of LOF variants in *APOC3* [31], encoding small apoC-III proteins that are associated with a low-risk lipid profile and fewer cardiovascular events [30,32], has emerged into trials with antisense oligonucleotides and are now approved in Europe for treating familial chylomicron syndrome patients with severe risk for pancreatitis. However, as of last year, the drug was deferred in children due to side effects such as thrombocytopenia [81,82]. At the moment, it is also not approved by the American FDA due to side effects. 

Furthermore, other targeted therapies currently on the market include inhibitors against microsomal transport protein of triglycerides (MTTP) and APOB (mipomersen) for cases that do not reach their target LDL. 

## 10. Conclusions

Successful genetic screening for FH and other dyslipidemias identifies many individuals each year who are at risk of cardiovascular disease. By identifying the underlying genetic cause, these individuals can receive the best treatment and cascade screening can identify additional family members in whom events can be prevented. Besides identifying these individuals, knowing the architecture of disease pathology provides opportunities for effective targeted drug development and individualized therapy and teaches us about new mechanisms for disease pathology.

## Figures and Tables

**Figure 1 genes-12-01168-f001:**
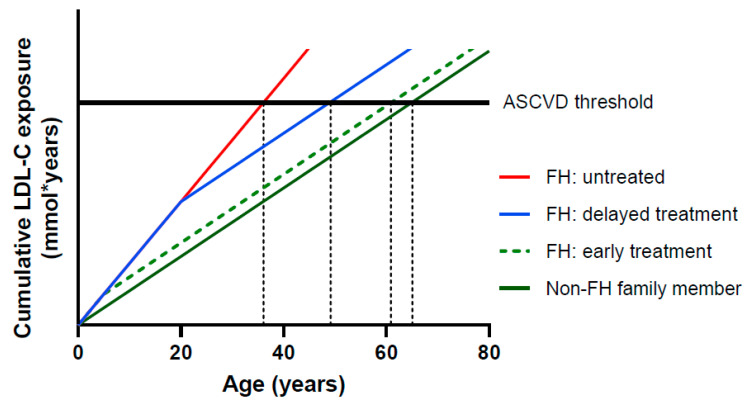
Relationship between cumulative LDL-C exposure and age. The horizontal black line represents a theoretical threshold of the cumulative LDL exposure required for development of atherosclerotic cardiovascular disease (ASCVD) [15], adapted from [16].

**Figure 2 genes-12-01168-f002:**
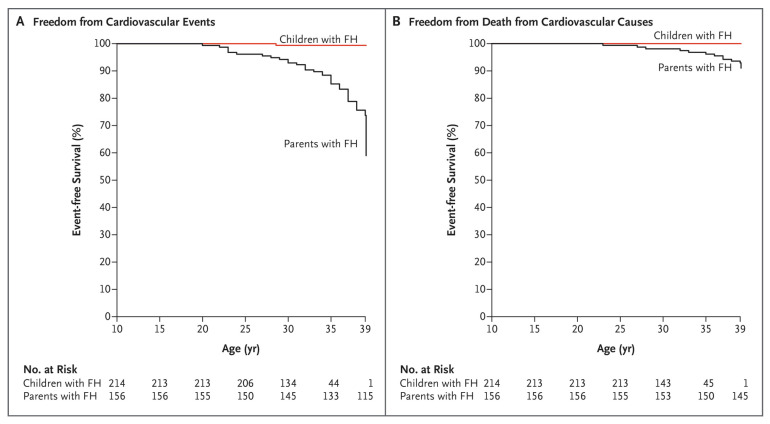
Kaplan–Meier curves for Patients with FH who began receiving statin therapy during childhood and their affected parents for whom statins were available much later in life. (**A**) Freedom from cardiovascular events; (**B**) freedom from death from cardiovascular causes [17].

**Figure 3 genes-12-01168-f003:**
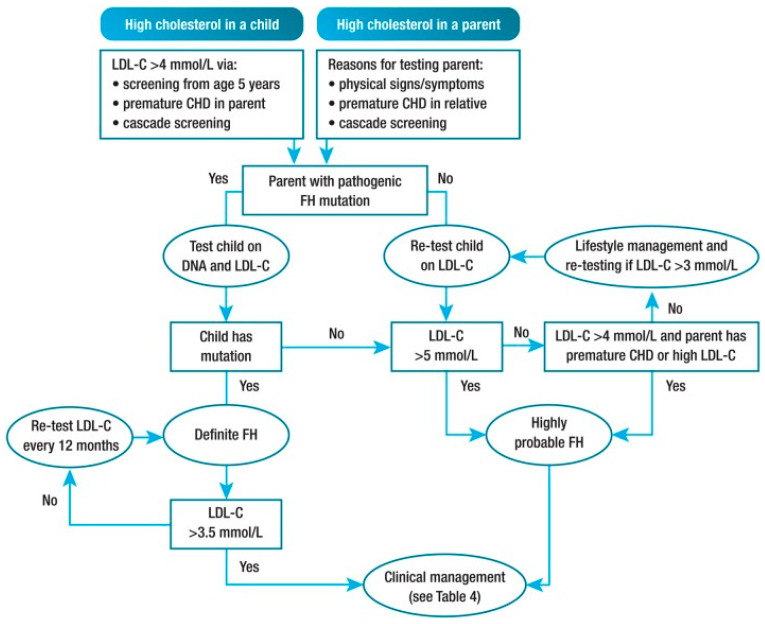
Potential strategy for diagnosis of FH in children and adolescents. CHD, coronary heart disease; FH, familial hypercholesterolemia; LDL-C, low-density lipoprotein cholesterol. Definitions: premature CHD is defined as a coronary event before age 55 years in men and age 60 years in women. Definite FH is defined as genetic confirmation of at least one FH-causing genetic variant. Close relative is defined as 1st or 2nd degree. Highly probable FH is based on clinical presentation (i.e., phenotypic FH), either an elevated LDL-c level ≥ 5 mmol/L in a child after dietary intervention or an LDL-c level ≥ 4 mmol/L in a child with a family history of premature CHD in close relatives and/or baseline high cholesterol in one parent. Cascade screening from an index case with an FH-causing variant may identify a child with elevated LDL-c levels ≥ 3.5 mmol/L [9].

**Table 1 genes-12-01168-t001:** Gene panel for NGS analysis of heritable dyslipidemias in the Netherlands. The panel is composed of 29 genes, of which 3 were FH-causing and 24 were causal for other dyslipidemias, and cover 10 phenotypes.

Phenotype/Disorder	Gene	Symbol	Ref Seq	Remarks
autosomal dominant hypercholesterolemia	low-density lipoprotein receptor	LDLR	NM_000527.4	dominant LOF variants
	apolipoprotein B	APOB	NM_000384.2	dominant LOF variants in exon 26 or 29; familial defective APOB-100
	proprotein convertase subtilisin/kexin type 9	PCSK9	NM_174936.3	dominant GOF variants
autosomal recessive hypercholesterolemia	LDL-receptor adaptor protein-1	LDLRAP1	NM_015627.2	recessive LOF variants
	lysosomal acid lipase	LIPA	NM_000235.3	recessive LOF; Wolman disease/cholesterolester storage disease
	ATP-binding cassette G5	ABCG5	NM_022436.2	recessive LOF variants; sitosterolemia
	ATP-binding cassette G8	ABCG8	NM_022437.2	recessive LOF variants; sitosterolemia
hypolipoproteinemia	apolipoprotein B	APOB	NM_000384.2	dominant LOF variants first halve of gene (gene dosage effect): hypobetalipoproteinemia
	proprotein convertase subtilisin/kexin type 9	PCSK9	NM_174936.3	dominant LOF variants; hypocholesterolemia
	angiopoietin-like 3	ANGPTL3	NM_014495.3	dominant LOF variants (gene dosage effect); combined hypolipidemia
	apolipoprotein C3	APOC3	NM_000040.1	dominant LOF variants
	microsomal triglyceride transfer protein	MTP	NM_000253.3	recessive LOF variants; abetalipoproteinemia
	inducible degrader of the LDL-receptor	IDOL	NM_013262.3	alias: *MYLIP,* dominant LOF variants
hypertriglyceridemia	lipoprotein lipase	LPL	NM_000237.2	dominant LOF variants, heterozygous need provoked by life style
	apolipoprotein C2	APOC2	NM_000483.4	recessive LOF variants
	apolipoprotein A5	APOA5	NM_052968.4	dominant LOF variants
	GPI anchored HDL binding protein 1	GPIHBP1	NM_178172.5	recessive LOF variants
	lipase maturation factor 1	LMF1	NM_022773.2	recessive LOF variants
	glycerol-3-phosphate dehydrogenase-1	GPD1	NM_005276.4	recessive LOF variants; transient infantile
dysbetalipoproteinemia	apolipoprotein E	APOE	NM_000041.3	recessive and dominant variants; need provoked by life style
hypoalphalipoproteinemia	ATP-binding cassette A1	ABCA1	NM_005502.3	LOF variants dominant: hypoalphalipoproteinemia; probably not fully penetrant, recessive: Tangier disease
	lecithin-cholesterol acyltransferase	LCAT	NM_000229.1	LOF variants dominant: hypoalphalipoproteinemia, recessive: Fish Eye disease
	apolipoprotein A1	APOA1	NM_000039.1	dominant LOF variants
hyperalphalipoproteinemia	scavenger receptor B1	SCARB1	NM_005505.4	dominant LOF variants
	cholesteryl ester transfer protein	CETP	NM_000078.2	dominant LOF variants
	lipase G	LIPG	NM_006033.3	dominant LOF variants, alias: endothelial lipase
	lipase C	LIPC	NM_000236.2	dominant LOF variants, alias: hepatic lipase, can also increase TG
cerebrotendinous xanthomatosis (CTX)	cytochrome P450, family 27A, polypeptide 1	CYP27A1	NM_000784.3	recessive LOF variants
chylomicron retention disease	secretion associated Ras related GTPase 1B	SAR1B	NM_016103.3	recessive LOF variants; Anderson disease
drug response	cytochrome P450, family 7A, polypeptide 1	CYP7A1	NM_000780.3	recessive LOF variants; statin resistance
	solute carrier organic anion transporter 1B1	SLCO1B1	NM_006446.4	dominant LOF variants; statin intolerance/decreased clearance

## Data Availability

Not applicable.

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
