# Peer review of "Successful Genetic Screening and Creating Awareness of Familial Hypercholesterolemia and Other Heritable Dyslipidemias in the Netherlands"

_genes, 2021, doi:10.3390/genes12081168_

Round 1
Reviewer 1 Report
The work present paper makes a proper review of the most current data of the FH. The authors also explain the strategy of the "genetic screening program for familial hypercholesterolemia (FH)" in the Netherlands, which has shown such good results since 1994. It is also important that they detail the candidate genes with mutations included in the program. In short, it is an useful descriptive study regarding the screening diagnosis of FH.
Please correct the following words
cohort of LDLR/APOB/PCSK9-negatieve (negative) hypercholesterolemia cases with normal triglyc-302
2. (S) statistics, E. Cardiovascular diseases statistics. 2020. 440
Author Response
Reviewer 1
We thank the reviewer for reading our manuscript with interest. The mentioned spelling errors are now corrected.

Reviewer 2 Report
In this review article, the importance of genetic testing is stressed by the analysis of a Dutch program of genetic testing for familial hypercholesterolemia. Albeit this point is relevant, as well as the movement from government funding to community funding, some aspects of the program results should be emphasized.
First, and to appeal to a large number of readers, the relevance of the genes tested should be stressed. It would be important that figures showing the relevance of these genes on cholesterol metabolism were included (for example metabolic pathways where these genes are involved). Also, since some of the more new and relevant findings in the program relay in the analysis of PCSK9 GOF, a figure showing the relevance of these mutations in the functions of the gene should be included.
Also, since the program has been carried out for several years, it should be of interest the inclusion of tables where the percentage and relevance of the different mutations that the program has detected on the Dutch population.
In conclusion, it is an interesting review, but the scientific findings of the screening program should be expanded. Also, the inclusion of figures that point out the relevance of the genes tested on cholesterol metabolism would make the review interesting for a broader population of readers.
Author Response
Reviewer 2
We thank the reviewer for reading our manuscript and raising interesting points of adding data of a more thoroughly executed analysis of 5 years of NGS. We agree this would indeed be very valuable additional information for the reader, however, our diagnostic database is not built for data analysis. Unfortunately, it is extremely time consuming to extract all these data from our database and perform the analysis. Nevertheless, we are planning to publish this in a future publication in which there is also more space for these extensive analyzes. This holds also true for the analysis of all PCSK9 variants we have found so far. Nevertheless, we agree with the author that this paragraph was a bit short in the current manuscript and extended this with a small paragraph in chapter 5. .
The reviewer is right that explaining the cholesterol metabolism is more appealing and clarifying for the reader and we therefore inserted a description of the cholesterol metabolism, as suggested, including most important genes included in our NGS panel. We restricted to those genes to keep focus on the genetic part of the article.

Reviewer 3 Report
The paper introduced the genetic screening for FH and other dyslipidemias in Netherlands. The following comments should be addressed to increase the value of the paper. Overall text can be reedited.
- The screening system should be concretely described in detail. For instance, which professionals (eg, general phycisions and specialists or nurses) are involved in the system? Or has the system been realized as the system of the community or health check-up? How is the awareness of the system at the community level? How is the advocation or announcement for the community?
- Are the ethics and law associated with the system?
- The outcome and your data of results by screening, up to date, should be concretely described in detail.
- How about the rejection/refuse rates of general people or children?
- The system can be more compared to the other countries.
- Is there any rural-urban difference in the system?
- Is there any genetic deviation within the country?
- The rationale to select other 24 genes can be more described in detail (expert concern, evidence, …).
- Title should include ‘Netherlands’.
- Abstract: row 9, that can be changed to which.
- Abstract, here and there in the text: genes can be expressed in an italic style.
- Row 47, -high must be corrected to -low.
- Row 79 , here and there in the text: review the use of however (adverb).
- Row 380-2, some refences are needed.
- Row 386, some refences are needed.
- Keywords were missing.
- The needed comments (COI etc.) were missing.
Author Response
Reviewer 3
We thank the reviewer for reading our manuscript and the interesting suggested points to increase the value of the paper.
1./2. Although extensively discussed in other articles, we agree with the reviewer to insert extra information about screening system, ethics and law associated with the system. Besides referring to an extra article, we added five sentences to paragraph 4 (The Dutch FH screening project).
- We agree with the reviewer and explain in two sentences the philosophy behind the calculated twenty years of funding by the government. We add three sentences about the actual outcome and the achievement of the original goal. And in two extra sentences, we show why the program was by far not finished.
- The rejection and refuse rates are mentioned in reference Umans-Eckenhausen (5-year follow-up) and reference Besseling (20-year follow-up)
5./6./7. Comparison of systems has been published by Kusters et al in Archives of Diseases in Childhood 2012. And in the Netherlands, with genetic field workers, it turned out not to be of any difference in the system between urban and rural. There was of course a difference between intensively cascade screened regions and less intensively screened regions. We discovered 1000 different mutations in the cascade screening, responsible for 80% of all mutations. Indeed, there are specific regions with geographical distribution of the most prevalent mutations, Kusters et al published about in the Neth Heart J in 2011: Founder mutations in the Netherlands. We decided not to mention these three points 5./6./7. in the present manuscript.
- A detailed description of the cholesterol metabolism containing most important genes of our screening have been added.
9. “the Netherlands” is now included in the title
10-11. Words/style changed
12. Low is correct. We screen for variants that cause high levels of lipoproteins and for variants causing low levels of lipoproteins.
13. Word “however” checked and changed when necessary
14-15. Reference added:
- E.A. Brinton, P.N. Hopkins, R.A. Hegele, et al. The association between hypercholesterolemia and sitosterolemia, and report of a sitosterolemia kindred. J Clin Lipidol, 12 (2018), p152-161
- Fong V, et al. Recent advances in ABCG5 and ABCG8 variants. Curr Opin Lipidol. 2021 Apr 1;32(2):117-122
- Wintjens R. et al. Global molecular analysis and APOE mutations in a cohort of autosomal dominant hypercholesterolemia patients in France. J Lipid Res. 2016 Mar;57(3):482-91
16. Key words have been added (Dyslipidemia, familial hypercholesterolemia, cholesterol, lipids, genetic screening)
17. Conflicts of interest and contributions are now added at the end of the manuscript.

Round 2
Reviewer 2 Report
The new version of the manuscript is improved. The focus on cholesterol metabolism and the role of the genes tested on the pathology has put in value the review. Also, the manuscript has been improved by adding the responses to the other reviewers. The new version is now directed to a broader population of readers. Still, it would be advisable that the database would have allowed a more detailed analysis, but the data presented is sufficient.
Reviewer 3 Report
There were no comments.